# Structural Model of Retention Intention of Nurses in Small- and Medium-Sized Hospitals: Based on Herzberg’s Motivation-Hygiene Theory

**DOI:** 10.3390/healthcare10030502

**Published:** 2022-03-09

**Authors:** Joo Yeon Lee, Mi Hyang Lee

**Affiliations:** 1Department of Nursing, Chungbuk Health & Science University, Cheongju 28150, Korea; wndus@chsu.ac.kr; 2Department of Nursing, Konyang University, Daejeon 35365, Korea

**Keywords:** Herzberg’s motivation, nurse, retention intention

## Abstract

The purpose of this study is to identify factors affecting the retention intention of nurses in small- and medium-sized hospitals and to perform a structural equation model study. Survey data of 348 nurses from 6 small and medium hospitals were analyzed. The collected data were analyzed using the SPSS 25.0 and the AMOS 25.0 programs. As a result of the study, it was confirmed that the endogenous variables influencing job satisfaction were calling, resilience, workplace bullying and nursing work environment, while resilience was the strongest variable as a factor influencing the nursing work environment. It was confirmed that the endogenous variables influencing intention to stay were calling, resilience, workplace bullying and job satisfaction, while job satisfaction was the strongest variable influencing intention to stay. To increase the retention intention of nurses in small and medium hospitals, it is necessary to provide measures to increase the value and meaning of work, and to increase resilience to overcome adversity and adapt to the circumstances. In addition, it is necessary to secure and maintain the resources of nurses in small- and medium-sized hospitals with a strategy to reduce workplace bullying and enhance job satisfaction by improving the organizational culture.

## 1. Introduction

With the increase in demand for nursing services caused by the diversification of healthcare policies, acquiring a sufficient nursing labor force is a key factor that increases patient satisfaction and allows quality nursing services [1]. According to a national nurse activity condition investigation, the average employment rate of those with a degree in nursing is around 70%, with around 45.5% of licensed nurses active in healthcare facilities [2]. As a result, shortages in the nursing labor force continue to occur in medical care facilities, and this is particularly serious in small- to mid-sized hospitals and local regions [3].

The biggest reason small- to mid-sized hospitals suffer from a nursing labor force shortage is that nurses prefer hospitals in the metropolitan area or general hospitals with better welfare and salary levels and thus transfer to them [4]. The nursing labor force shortage lowers the quality of nursing services and patient satisfaction while also increasing the workload of other nurses, resulting in a vicious cycle of nurse turnover [5] as well as negative effects such as patient safety accidents, including medication errors, bedsores, falling accidents, etc. [6].

While research related to turnover and turnover intention has been actively carried out to resolve the issue of national shortages in the nursing labor force, there has recently been more interest in the intention to stay, or nurses wanting to remain in an organization [7]. Identifying the factors that affect a nurse’s satisfaction with the current organization and ultimately their intention to stay is also very meaningful from a nursing labor force management perspective. Thus, strengthening the causes that can increase the intention to stay can help to acquire competent nurses and enhance the effectiveness of the organization, so identifying the intention to stay is more important than turnover intention [8]. While the causal factors for intention to stay and for turnover are similar, distinct variables are being suggested. Salary, promotion and work environment are important factors for turnover intention [5], whereas individual motivation is the factor that largely affects the intention to stay [9].

Motivation is the will of the individual to put in their best efforts while working to achieve the goals of the organization, and it has a positive impact on the intention to stay [10]. Herzberg’s motivation-hygiene theory provides the member with motivation about work and interest in the work environment—motivators are satisfiers such as an individual’s effort, belief, growth probability, etc., and dissatisfiers are hygiene factors such as interpersonal relationships, positions, work requirements, etc. [11]. The important message suggested by motivation-hygiene theory is that strengthening motivators is essential for manpower management in the organization because they are important variables for the intention to stay, and that the dissatisfaction of members can be reduced by improving the dissatisfiers, or the work environment.

Looking at preceding research on the intention to stay targeting small- to mid-sized hospital nurses, there have been studies on stress coping abilities, the effect of resilience on the intention to stay [4], impact of the nursing work environment and care-consideration between colleagues on the intention to stay [12], factors affecting the intention to stay [13,14], etc. However, it was difficult to find research that identified the intention to stay and established a structural model to explore the factors that cause the motivation for small- to mid-sized hospital nurses to stay in the organization.

Therefore, based on Herzberg’s motivation-hygiene theory, this study seeks to establish a hypothetical model on the variables that affect the intention to stay for small- and mid-sized hospital nurses based on a literature review of relevant preceding research that explains this. Additionally, this study aims to verify the validity of the model to prepare the grounds for basic information that can be used to establish a nursing labor force management solution to enhance the intention to stay for small- to mid-sized hospital nurses.

### Conceptual Basis and Hypothetical Model

This study established a conceptual basis by combining the factors affecting the intention to stay for small- to mid-sized hospital nurses based on Herzberg’s motivation-hygiene theory. Motivators are individual factors referred to as work satisfiers that enhance an individual’s growth and potential as well as devotion to work, whereas hygiene factors are work dissatisfiers such as interpersonal relationships, working conditions, etc. [11]. Therefore, this study has identified the motivators as calling and resilience, and the hygiene factors as leader–member exchange relationship, workplace bullying and nursing work environment.

The motivator factor calling is the attitude and belief towards work, which is the main variable that leads to the formation of goals [15]. It was found that the higher the calling of the nurse, the higher the intention to stay [16]. Resilience is the social-psychological factor that allows adjustment and growth amidst adversities [17]. Resilience increases stress coping ability and intention to stay [4]. In addition, the higher the resilience of a nurse, the higher the job satisfaction [18].

The leader–member exchange relationship hygiene factor is an interpersonal relationship characteristic where amicable relationships in the organization enhance systematic activity and work performance [19]. The higher the leader–member exchange relationship, the lower the turnover intention and the higher the organizational commitment to job satisfaction [20]. Workplace bullying refers to applying physical and mental pain to other workers exceeding the appropriate range in work, where more workplace bullying results in lower work performance [21]. The more workplace bulling, the lower the intention to stay and the lower the organizational commitment [22]. The nursing work environment is not only related to cooperative relationships, organizational culture, and the work environment atmosphere, but also plays a big role in maintaining the nursing labor force [23]. The better the teamwork of nurses in the nursing work environment, the higher the job satisfaction and the lower the burnout and turnover intention [24].

Therefore, as opposed to eliminating dissatisfactions, this study sought to identify the work satisfiers and increase the intention to stay for small- to mid-sized hospital nurses for labor force management by modifying the motivation-hygiene theory suggested by Herzberg to prepare a conceptual basis for the study.

## 2. Materials and Methods

### 2.1. Study Design

To predict the variables that explain the intention to stay for small- to mid-sized hospital nurses, this study established a theoretical model for factors affecting the intention to stay based on the conceptual basis of Herzberg’s motivation-hygiene theory and a review of preceding research and performed a structural equation model study to verify the appropriateness of the model and the hypothesis using the collected data.

### 2.2. Subjects

The subjects of this study were nurses working in small- to mid-sized hospitals in Korea with 30 to 300 beds. Convenience sampling was performed for small- to mid-sized hospitals in Korea to select nurses working in a total of six hospitals, two in the Gyeonggi area, two in the Daejeon area and two in the Gyeongbuk area. This study collected data from July to August 2019. Structural equation studies require at least 200 samples, and while there is no absolute sample size, between 200 and 400 is considered appropriate [25]. Therefore, this study distributed 370 questionnaires and recovered 360 copies (recovery rate 97%), then eliminated 12 questionnaires through data review for incomplete or duplicate answers (removal rate 3.4%) to finally analyze a total of 348 questionnaires (94%).

### 2.3. Measures

#### 2.3.1. Calling

Calling is a tool developed by Dik et al. [15] and amended/supplemented by Sim and Yoo [26], and was used after receiving consent from the authors. The tool consists of three subareas, transcendental calling, purpose/meaning and pro-social orientation, where a total of 12 questions are answered based on a 4-point Likert scale, with 1 point meaning “not applicable” and 4 points meaning “fully applicable”. Higher scores indicate a higher sense of calling. In the study by Dik et al. [15], the Cronbach’s alpha was 0.85, while in this study it was 0.86.

#### 2.3.2. Resilience

Resilience is a tool developed by Connor and Davidson [17] and was used after consent from the authors as well as payment for the use of the tool. The tool consists of 5 subareas: tenacity, endurance, optimism, support and spirituality, with a total of 25 questions, answered based on a 5-point Likert scale, with 0 points meaning “never” and 4 points meaning “almost always true”. Higher scores indicate higher resilience. The Cronbach’s alpha in the study by Connor and Davidson [17] was 0.89, while in this study it was 0.93.

#### 2.3.3. Leader–Member Exchange

The leader–member exchange relationship is a tool developed by Liden and Madyln [27] and was used after gaining consent from the writers. The tool consists of 4 subareas: emotional connectedness, loyalty, contribution and respect for professionalism, with a total of 12 questions answered based on a 5-point Likert scale, with 1 point meaning “not true” and 5 points meaning “very true”. Higher scores indicate higher leader–member exchange relationships. The Cronbach’s alpha in the study by Liden and Madyln [27] was 0.89, while in this study it was 0.95.

#### 2.3.4. Workplace Bullying

Workplace bullying is a tool developed by Lee et al. [21] to fit real domestic situations and was used after gaining consent from the writers. The tool consists of 3 subareas: verbal violence, work-related violence and physical violence, with a total of 16 questions answered based on a 4-point Likert scale, with 1 point meaning “not true” and 4 points meaning “very true”. Higher scores indicate more exposure to workplace bullying. The Cronbach’s alpha in the study by Lee et al. [21] was 0.90, while in this study it was 0.93.

#### 2.3.5. Nursing Work Environment

Nursing work environment is a tool developed by Lake [28] and amended/adapted by Cho et al. [23] and was used after gaining consent from the writers. The tool consists of five subareas: nurse participation in hospital operation, work basis for quality nursing, nurse manager’s leadership/ability and support for nurses, sufficient labor force and material resources and cooperation between nurses and doctors. The tool has a total of 29 questions answered based on a 4-point Likert scale, with 1 point meaning “not true” and 4 points meaning “very true”. Higher scores indicate more positive perceptions about the work environment. The Cronbach’s alpha in the study by Lake [28] was 0.82, while in this study it was 0.93.

#### 2.3.6. Job Satisfaction

Work satisfaction used the Visual Analog Scale (VAS) to evaluate the subjective work satisfaction for small- to mid-sized hospital nurses. The tool uses a 10-point VAS, where the degree of work satisfaction ranges from 1 point, or “not true”, to 10 points, or “very true”. Higher scores indicate greater job satisfaction.

#### 2.3.7. Intention to Stay

Intention to stay is a tool developed by Cowin [29] and adapted by Kim [30] and was used after gaining consent from the authors. The tool has a total of 6 questions answered based on an 8-point Likert scale, with 1 point meaning “not true” and 8 points meaning “very true”. Higher scores indicate a higher intention to stay. The Cronbach’s alpha in the study by Cowin [29] was 0.97, while in this study it was 0.91.

### 2.4. Data Collection

This study is a self-recording questionnaire survey and received approval from K university’s Bioethics Committee (IRB No. KUH-2019-276-01). Data collection was performed from 15 July to 30 August 2019. To distribute the questionnaires, we visited the small- to mid-sized hospitals or used email to submit the Bioethics Committee result notice, research plan and research description to obtain approval for the study from each hospital. The organizations for the study were from six locations (two in Gyeonggi, two in Gyeongbuk and two in Daejeon), selected by convenience sampling, and the subjects were nurses who voluntarily provided consent in writing after receiving explanations about the purpose, method and procedure of the study. Additionally, we explained to the participants about the anonymity of the research data, the guarantee of privacy and the option to opt out of the study at any time without any disadvantages. All questionnaires were recovered in individual envelopes that could be enclosed, and the nurses who participated in the study were given small presents in return.

### 2.5. Data Analysis

The collected data were analyzed using SPSS WIN/25.0 (IBM, Armonk, NY, USA) and AMOS version 25.0 programs (IBM, Armonk, NY, USA). The general characteristics of the subjects and the descriptive statistics for each research variable were analyzed with frequency analysis and descriptive statistics. Measure of skewness and kurtosis were reviewed to verify the data’s normality, and the multicollinearity between the measured variables was calculated using Pearson’s correlation coefficient. Confirmatory factor analysis was used to analyze the validity of the measuring tool and covariance structural analysis with the maximum likelihood method was performed to verify the hypothesis and evaluate the appropriateness of the hypothetical model. The goodness-of-fit test was analyzed using the χ^2^ value, goodness-of-fit index (GFI), adjusted goodness-of-fit index (AGFI), Tucker–Lewis index (TLI), normed fit index (NFI), comparative fit index (CFI), root mean square error of approximation (RMSEA) and standardized root mean residual (SRMR). The bootstrapping method was used for statistical significance testing on the direct, indirect and overall effects of the hypothetical model.

## 3. Results

### 3.1. General Characteristics of the Subjects

The demographic characteristics of the subjects of this study were that they were mainly females (318, 91.4%), with the highest number in the 20~29 age group (153, 44.0%) and an average age of 33.47 ± 9.17. The highest proportion in terms of marital status was single (190, 54.6%), and bachelor’s degree (194, 55.8%) in terms of education. For their clinical career in the current hospital, the highest number was 1 to 3 years (96, 27.6%), with a total average of 6.54 ± 7.44 years, and for the total clinical career, the highest number was more than 10 years (142, 40.8%), with an average of 9.24 ± 8.17 years. The highest proportions of other characteristics include general nurse for title (317, 91.1%), medical ward for department (96, 27.6%) and three-shift system for work pattern (227, 65.2%) (Table 1).

### 3.2. Descriptive Statistics and Normality Test for the Measured Variable

The descriptive statistics for variables used in the hypothetical model of this study are as follows (Table 2). Upon checking the measure of skewness and kurtosis to check the normal distribution of the variables in this study, all variables did not exceed the absolute value of 2 to satisfy normality. Correlation analysis of the measured variables to test for multicollinearity resulted in values of 0.00~0.79, and while multicollinearity can be suspected if the correlation between variables is higher than 0.8, this was not the case in this study, thus eliminating any issues with multicollinearity. With confirmatory factor analysis to test the validity of the variables, the average variance extracted (AVE) value was greater than 0.5 to satisfy the discriminant validity criterion. Construct reliability (CR) was used to evaluate convergent validity, and the CR satisfying standard was greater than 0.7 for all variables, thus also satisfying convergent validity. In confirmatory factor analysis, the factor loading size must be 0.5 or greater to be considered appropriate, and upon analysis of the standardized index, spirituality, which is one of the measured variables for resilience, had a standardized index value of 0.47. Thus, spirituality was removed when the hypothesis model was verified.

### 3.3. Hypothesis Model Test

#### 3.3.1. Goodness-of-Fit Test for the Hypothesis Model

To evaluate the fit of the hypothesis model and the relationship between factors in this study, the structural equation model was analyzed with maximum likelihood estimation. The goodness-of-fit for the hypothesis model was χ^2^ = 450.135 (df = 170, *p* < *0*.001), normed χ^2^ = 2.648, GFI = 0.891, AGFI = 0.852, NFI = 0.895, TLI = 0.915, CFI = 0.931, RMSEA = 0.069, and SRMR = 0.063. While the GFI and NFI values of this study were slightly below the standards, the overall model had values similar to an optimal model. Therefore, the hypothesis model was selected after deciding that the goodness-of-fit of the overall model satisfied the recommended standards.

#### 3.3.2. Correlation between Measured Variables and Multicollinearity

Table 3 summarizes the results of analyzing the correlation between the measured variables in this study. The correlation between the measured variables was 0.00 to 0.79, which is less than 0.80 and confirms no multicollinearity between the measured variables.

#### 3.3.3. Discriminant Validity

In this study, the average variance extracted (AVE) value of each latent variable was more significant than the coefficient of determination (r^2^) of each latent variable, indicating that the discriminant validity was satisfied (Table 4).

#### 3.3.4. Verifying Effectiveness of the Hypothesis Model

The final model of this study consists of eleven hypotheses, and the analysis revealed that eight of the hypotheses showed statistically meaningful impacts. Work satisfaction, the endogenous variable for small- to mid-sized hospital nurses, showed that calling and resilience among the motivators and workplace bullying and nursing work environment among hygiene factors had effects, with an explanatory power of 28.8%. Among these, resilience was identified as the variable with the greatest impact on work satisfaction (β = 0.24, CR = 3.17, *p* = 0.002). For the intention to stay, calling (motivator), workplace bullying (hygiene factor) and work satisfaction had effects, with an explanatory power of 37.8%. Among these, work satisfaction was identified as the variable with the greatest impact on the intention to stay (β = 0.37, CR = 5.66, *p* < 0.001) (Table 5).

#### 3.3.5. Effectiveness Analysis of the Final Model

The analysis of direct, indirect and overall effects of the hypothesis model in this study was as follows (Table 3). Out of a total of eleven routes, there were eight meaningful routes considering the direct, indirect and overall effects (Figure 1). Calling, resilience, workplace bullying and nursing work environment had direct effects on work satisfaction, whereas workplace bullying and work satisfaction had direct effects on the intention to stay.

## 4. Discussion

This study aimed to apply Herzberg’s motivation-hygiene theory and deduce a hypothetical model to predict the intention to stay for small- to mid-sized hospital nurses.

In terms of the general characteristics of small- to mid-sized hospital nurses, higher age, more than 10 years of clinical experience in the current company, less than 1 year or more than 10 years of total clinical experience, higher positions and full-time nurses had a higher intention to stay. The research by Yom, Yang and Han [31] showed higher intention to stay for those over 40 and with current/total clinical experience of more than 10 years, similar to the results of this study. This is because more experience increases the adaptability with work and the organization and makes it difficult to move to other jobs. Research by Yu, Kang, Yu and Park [32] showed high intention to stay for nurses with less than 1 year of work experience, which is considered to be because new nurses have recently joined and thus have the will to adjust to the new job. However, the average number of nurses who participated in this study was 33.47 years old, with 73% between 20 and 39 years old, and 6.9% over 50 years old. In the future, repeated studies are needed on the analysis of differences in employment intentions according to age with a similar composition by age. In addition, women account for more than 90% of domestic nurses, so there are limitations in analyzing gender differences.

The analysis in this study to explain the work satisfaction of small- to mid-sized hospital nurses resulted in calling, resilience, workplace bullying and nursing work environment, with the nursing work environment having the greatest impact on work satisfaction. The internal motivation of value and satisfaction about work has a greater impact on work satisfaction than external motivations such as salary, compensation, etc. [11]. As stated in Herzberg’s motivation-hygiene theory, the motivators must be strengthened, and the hygiene factors must be managed to reduce work dissatisfaction and enhance work satisfaction for small- to mid-sized hospital nurses. Therefore, organizations must identify factors to enhance work satisfaction for nurses by internal marketing and implement systematic and consistent policies.

The analysis in this study to explain the intention to stay for small- to mid-sized hospital nurses resulted in calling, resilience, workplace bullying and work satisfaction as variables, with work satisfaction having the greatest impact on the intention to stay. This result was very similar to preceding research [14,16]. This shows that the internal factor (work satisfaction) is a more important factor in the intention to stay for small- to mid-sized hospital nurses than the external environment. Calling not only strengthens the internal motivation to adjust to the rapidly changing medical environment and examine the abilities/knowledge related to work, but also results in a higher evaluation of the occupational value of nursing to increase the intention to stay. Additionally, higher resilience resulted in a higher intention to stay, which is thought to be because a positive attitude allows active responses to stress and better adjustment to the organization; thus, nurses with high resilience also have a high intention to stay. Therefore, resilience is a required psychology for nurses who need to make immediate decisions while taking care of patients with various medical requirements.

A study by Kim and Park [13] showed that workplace bullying affects the intention to stay for small- to mid-sized hospital nurses, matching the results of this study. Workplace bullying in the nursing environment reduces the time and luxury for the senior nurse to educate and take an interest in nurses with less experience, and berating by senior nurses negatively affects the ability of new nurses to adjust to the organization, lowering the intention to stay. Training new nurses while also performing their duties can be burdensome on experienced nurses. This can cause physical/mental stress for experienced nurses. Therefore, to increase the intention to stay, measures for emotional exhaustion are needed, as well as emotional support for both new and experienced nurses, and there needs to be a nurse training system to educate new nurses or those with low work adaptability.

The leader–member exchange relationship and nursing work environment did not have any impact on the intention to stay, supporting Herzberg’s motivation-hygiene theory. Work satisfaction had an impact on the intention to stay, matching the results of a study carried out by Bae et al. [4] targeting small- to mid-sized hospital nurses. Additionally, since work satisfaction is the strongest variable in increasing the intention to stay, strategies to enhance the work satisfaction of small- to mid-sized hospital nurses would be very effective from a labor force management perspective.

The biggest reason nurses remain and work in an organization is that they consider the job one that helps patients and guardians [14]. This means that nurses work with a sense of service and duty as professionals. While enhancing the satisfaction with hygiene factors is also important, internal motivation for work in the subconscious of nurses must be increased to heighten their intention to stay. An equal level of authority and responsibility, and sufficient recognition of the work can help provide motivation [10], with such motivation in the members enhancing work performance [11]. On the other hand, while enhancing the hygiene factors can provide positive meaning to work for a short time, this is taken for granted as time passes, thus potentially losing the motivation to continue working [33]. Nurses must go beyond satisfaction with working conditions or the improvement in the work environment to develop abilities for individual growth, feel a sense of worth with nursing and have a professional perception of nursing duties. The strength of this study is meaningful in verifying Herzberg’s motivation-hygiene theory. The weakness is that most of the subjects of the study are women and are concentrated in their 20s and 30s, so there are limitations in generalization.

## 5. Conclusions

This study was carried out to apply Herzberg’s motivation-hygiene theory to establish a structural model of the exogenous variables of calling, resilience, leader–member exchange relationship, workplace bullying and nursing work environment, as well as the endogenous variables of work satisfaction and intention to stay.

As a result of this study, calling, resilience, work satisfaction and workplace bullying were identified as factors affecting the intention to stay for small- to mid-sized hospital nurses. Work satisfaction, which has the greatest impact on the intention to stay, was largely affected by the motivators calling and resilience. This means that from a long-term perspective, strengthening internal motivators as opposed to hygiene factors is required to increase work satisfaction and intention to stay for small- to mid-sized hospital nurses. Training to enhance calling and resilience must start from the time of nursing school so that nurses can adjust to the clinical scene and have positive perceptions. Additionally, studies are required to identify variables related to workplace bullying and prepare approaches to resolve this. Consistent research is needed to identify factors related to the intention to stay and to develop programs that increase this intention so that the tenure rate in the clinical field can be increased for efficient labor force management.

Based on the study results above, we would like to suggest the following. First, additional studies are required to verify Herzberg’s motivation-hygiene theory with nurses and to consider other variables that affect the intention to stay. Second, activating preceptorship training that fits the coping abilities of new nurses and positive organizational culture enhancement programs are required to enhance the intention to stay. Third, Herzberg’s motivation-hygiene theory must be expanded to general hospitals and large hospitals, etc., and the impact of the characteristics of each hospital organization on the intention to stay must be researched from a multilateral perspective. This is a regional study and has limitations in research.

## Figures and Tables

**Figure 1 healthcare-10-00502-f001:**
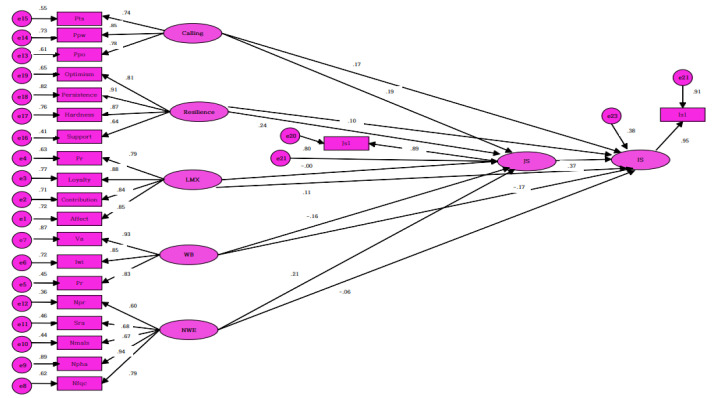
Path diagram for the hypothetical model. Calling: Pts = Presence—transcendent summons, Ppw = Presence—purposeful work, Ppo = Presence—prosocial orientation. Resilience: optimism, persistence, hardiness, support. LMX = Leader–member exchange. Pr = Professional respect, loyalty, contribution, affect. WB = Workplace bullying. Va = Verbal attacks, Iwi = Improper work instruction, Pt = Physical threats, NWE = nursing work environment, Npr = Nurse–physician relations, Sra = Staffing and resource adequacy, Nmals = Nurse manager ability, leadership and support, Npha = Nurse participation in hospital affairs, Nfqc = Nursing foundations for quality of care, JS = job satisfaction, IS = intention to stay.

**Table 1 healthcare-10-00502-t001:** General characteristics of participants (*n* = 348).

Variables	Categories	*n*	%	Mean ± SD
Gender	Male	30	8.6	
	Female	318	91.4	
Age (year)	20~29	153	44.0	33.47 ± 9.17
	30~39	101	29.0	
	40~49	70	20.1	
	≥50	24	6.9	
Marital status	Married	158	45.4	
	Single	190	54.6	
Education level	Diploma	140	40.2	
	Bachelor	194	55.8	
	≥Graduate	14	4.0	
Period working in the current department (years)	<1	70	20.1	6.54 ± 7.44
	1~<3	96	27.6	
	3~<5	41	11.8	
	5~<10	49	14.1	
	≥10	92	26.4	
Clinical career (years)	<1	32	9.2	9.24 ± 8.17
	1~<3	67	19.2	
	3~<5	43	12.4	
	5~<10	64	18.4	
	≥10	142	40.8	
Position	Staff nurse	317	91.1	
	≥Charge nurse	31	8.9	
Type of unit	Medical ward	96	27.6	
	Surgical ward	95	27.3	
	Special part	86	24.7	
	Comprehensive nursing care service ward	62	17.8	
	Outpatient	9	2.6	
Work shift	Three shifts	227	65.2	
	Two shifts	28	8.1	
	Full-time	78	22.4	
	Night shift	15	4.3	

**Table 2 healthcare-10-00502-t002:** Verification of normality of measurement variables (*n* = 348).

Latent Variable	Measured Variable	Min	Max	Mean ± SD	Skewness	Kurtosis	CR	AVE
Calling		1.17	3.75	2.17 ± 0.49	0.55	0.48	0.94	0.63
	Transcendental calling	1.00	4.00	1.93 ± 0.63	0.44	0.05		
	Purpose/Meaning	1.00	4.00	2.34 ± 0.55	0.34	0.27		
	Prosocial orientation	1.00	4.00	2.25 ± 0.52	0.09	0.19		
Resilience		0.81	3.97	2.58 ± 0.55	−0.03	0.04	0.95	0.66
	Hardness	0.67	3.89	2.36 ± 0.59	0.09	0.19		
	Patience	0.88	4.00	2.54 ± 0.58	0.13	0.07		
	Optimism	0.25	4.00	2.47 ± 0.70	−0.08	0.01		
	Support	1.00	4.00	2.93 ± 0.69	−0.48	−0.18		
	Spirituality	0.00	4.00	2.09 ± 0.77	0.07	−0.14		
Leader–member exchange		1.17	5.00	3.55 ± 0.61	0.01	0.29	0.95	0.71
	Affect	1.00	5.00	3.52 ± 0.68	0.04	0.27		
	Loyalty	1.67	5.00	3.52 ± 0.71	0.04	−0.12		
	Contribution	1.00	5.00	3.46 ± 0.66	−0.12	0.31		
	Professional Respect	1.00	5.00	3.71 ± 0.73	−0.24	0.11		
Workplace Bulling		1.00	3.42	2.57 ± 0.53	0.82	0.25	0.95	0.68
	Verbal attacks	1.00	4.00	1.87 ± 0.62	0.44	−0.36		
	Improper work instruction	1.00	3.75	1.83 ± 0.62	0.46	−0.37		
	Physical treat	1.00	3.50	1.37 ± 0.59	1.54	1.68		
Nursing Work Environment		1.59	3.88	2.57 ± 0.42	0.16	0.02	0.95	0.56
	Participation in hospital affairs	1.00	3.89	2.49 ± 0.50	−0.08	0.01		
	Nursing foundation for quality of care	1.44	4.00	2.65 ± 0.44	−0.14	0.15		
	Nursing manager ability, leader ship, and support	1.50	4.00	2.91 ± 0.50	−0.14	−0.08		
	Staffing and resource adequacy	1.00	3.75	2.24 ± 0.65	0.08	−0.67		
	Nurse-physician relations	1.00	4.00	2.58 ± 0.57	−0.57	0.52		
Job satisfaction		1.00	10.00	5.74 ± 1.90	−0.12	−0.52	0.82	0.80
Retention intention		1.00	8.00	5.21 ± 1.43	−0.21	−0.07	0.84	0.91

**Table 3 healthcare-10-00502-t003:** Correlation between measured variables.

	Calling			Resilience					LMX				Workplace Bulling				NWE				JS	IS
	1	2	3	4	5	6	7	8	9	10	11	12	13	14	15	16	17	18	19	20	21	22
1	1																					
2	0.62	1																				
3	0.64	0.65	1																			
4	0.28	0.45	0.33	1																		
5	0.32	0.54	0.39	0.79	1																	
6	0.32	0.46	0.33	0.72	0.71	1																
7	0.11	0.28	0.17	0.54	0.60	0.53	1															
8	0.45	0.29	0.26	0.41	0.38	0.47	0.24	1														
9	0.17	0.23	0.17	0.19	0.22	0.19	0.21	0.15	1													
10	0.18	0.21	0.21	0.22	0.25	0.21	0.23	0.20	0.74	1												
11	0.18	0.26	0.19	0.31	0.32	0.26	0.23	0.17	0.71	0.74	1											
12	0.08	0.14	0.12	0.17	0.23	0.15	0.17	0.10	0.68	0.69	0.67	1										
13	−0.04	−0.08	0.07	−0.11	−0.16	−0.11	−0.26	0.05	−0.25	−0.23	−0.24	−0.14	1									
14	0.00	−0.12	0.04	−0.12	−0.20	−0.17	−0.26	0.01	−0.21	−0.21	−0.23	−0.18	0.79	1								
15	0.08	−0.09	0.01	−0.13	−0.17	−0.13	−0.29	0.06	−0.13	−0.14	−0.14	−0.14	0.63	0.56	1							
16	0.14	0.16	0.16	0.17	0.18	0.15	0.13	0.46	0.50	0.46	0.47	0.47	−0.15	−0.19	−0.10	1						
17	0.18	0.15	0.21	0.13	0.13	0.15	0.07	0.19	0.03	0.13	0.11	0.06	−0.05	−0.17	0.09	0.41	1					
18	0.17	0.24	0.25	0.18	0.21	0.15	0.14	0.13	0.09	0.21	0.15	0.16	−0.06	−0.14	−0.03	0.39	0.45	1				
19	0.23	0.21	0.22	0.17	0.21	0.19	0.07	0.21	0.23	0.32	0.26	0.25	−0.04	−0.13	0.09	0.61	0.66	0.57	1			
20	0.16	0.15	0.16	0.11	0.14	0.14	0.09	0.18	0.22	0.24	0.19	0.25	−0.03	−0.05	0.03	0.58	0.49	0.45	0.75	1		
21	0.23	0.37	0.19	0.31	0.34	0.36	0.27	0.23	0.20	0.16	0.26	0.13	−0.17	−0.23	−0.12	0.25	0.32	0.21	0.27	0.20	1	
22	0.23	0.42	0.17	0.31	0.35	0.37	0.27	0.20	0.31	0.18	0.34	0.17	−0.27	−0.25	−0.23	0.23	0.11	0.09	0.16	0.16	0.45	1

LMX: Leader–member exchange, NWE: Nursing work environment, JS: Job satisfaction, IS: Intention to stay. 1: Presence—transcendent summons, 2: Presence—purposeful work, 3: Presence—prosocial orientation, 4: Hardiness, 5: Persistence, 6: Optimism, 7: Support, 8: Spirituality, 9: Professional respect, 10: Loyalty, 11: Contribution, 12: Affect, 13: Verbal attacks, 14: Improper work instruction, 15: Physical threats, 16: Nurse–physician relations, 17: Staffing and resource adequacy, 18: Nurse manager ability, leadership and support, 19: Nurse participation in hospital affairs, 20: Nursing foundation for quality of care, 21: Job satisfaction, 22: Intention to stay.

**Table 4 healthcare-10-00502-t004:** Discriminant validity.

Latent Variable	Calling	Resilience	LMX	WB	NWE	JS	IS
	r						
	r^2^						
	(*p*)						
Calling	0.63 ^†^						
resilience	0.44	0.66 ^†^					
	0.19						
	(<0.001)						
LMS	0.23	0.29	0.71 ^†^				
	0.05	0.08					
	(<0.001)	(<0.001)					
WB	−0.01	−0.24	−0.24	0.68 ^†^			
	0	0.08	0.06				
	–0.89	(<0.001)	(<0.001)				
NWE	0.27	0.21	0.33	−0.09	0.56 ^†^		
	0.07	0.05	0.11	0.01			
	(<0.001)	(<0.001)	(<0.0014)	−0.085			
JS	0.31	0.37	0.21	−0.20	0.32	0.80 ^†^	
	0.09	0.14	0.04	0.04	0.1		
	(<0.001)	(<0.001)	(<0.001)	(<0.001)	(<0.001)		
IS	0.31	0.38	0.28	−0.28	0.18	0.45	0.91 ^†^
	0.1	0.14	0.08	0.24	0.03	0.2	
	(<0.001)	(<0.001)	−0.001	(<0.001)	(<0.001)	(<0.001)	

LMX: Leader–member exchange, WB: Workplace bullying, NWE: Nursing work environment, JS: Job satisfaction, IS: Intention to stay. † Average variance extracted (AVE) value for each factor.

**Table 5 healthcare-10-00502-t005:** Effect verification of hypothetical models (*n* = 348).

Endogenous Variable	Exodoenous Variable	B	SE	β	CR (*t*)	*p*	Direct Effect (*p*)	Indirect Effect (*p*)	Total Effect (*p*)	SMC
Job satisfaction	Calling	0.53	0.21	0.19	2.46	0.014	0.19 (0.039)	−	0.19 (0.039)	0.288
	Recovery resilience	0.61	0.19	0.24	3.17	0.002	0.24 (0.007)	−	0.24 (0.007)	
	Leader–member exchange relationship	−0.00	0.13	−0.00	−0.02	0.983	−0.00 0.997	−	−0.00 (0.997)	
	Workplace harassment	−0.46	0.17	−0.16	−2.64	0.008	−0.16 (0.019)	−	−0.16 (0.019)	
	Nursing Work environment	0.68	0.20	0.21	3.42	<0.001	0.21 (0.018)	−	0.21 (0.018)	
Retention Intention	Calling	0.44	0.18	0.17	2.43	0.015	0.17 (0.106)	0.07 (0.039)	0.24 (0.023)	0.378
	Recovery resilience	0.23	0.16	0.10	1.43	0.154	0.10 (0.225)	0.09 (0.016)	0.19 (0.034)	
	Leader–member exchange relationship	0.20	0.11	0.11	1.88	0.060	0.11 (0.073)	−0.00 (0.996)	0.11 (0.081)	
	Workplace harassment	−0.46	0.16	−0.17	−3.12	0.002	−0.17 (0.011)	−0.06 (0.022)	−0.23 (0.016)	
	Nursing Work environment	−0.18	0.17	−0.06	−1.04	0.300	−0.06 (0.381)	0.08 (0.009)	0.02 (0.689)	
	Job satisfaction	0.35	0.06	0.37	5.66	<0.001	0.37 (0.021)	−	0.37 (0.021)	

SE: Standard error, CR: Construct reliability, SMC: Squared multiple correlation.

## Data Availability

The data presented in this study are available upon request from the corresponding author. The data are not publicly available due to data restriction policies.

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
