# Peer review of "Structural Model of Retention Intention of Nurses in Small- and Medium-Sized Hospitals: Based on Herzberg’s Motivation-Hygiene Theory"

_healthcare, 2022, doi:10.3390/healthcare10030502_

Round 1

Reviewer 1 Report

Dear Authors,
No question, you have completed an enormous job, so it was an honour to be among the ones to read your paper first.
Unfortunately, I could not comprehend all your messages due to strange word choices and sentence structures.
To improve this paper, I suggest following the author guide strictly and providing all the appropriate information required in all the sections.
You may want to re-structure the paper to support your conclusions, starting with a problem statement based on a wider literature review. Maybe you will find relevant some of the following documents:
https://www1.health.gov.au/internet/main/publishing.Nsf/content/29418ba17e67abc0ca257d9b00757d08/$file/nursing%20workforce%20sustainability%20-%20improving%20nurse%20retention%20and%20productivity%20report.Pdf
https://www.icn.ch/system/files/documents/2021-05/ICN%20Toolkit_2021_ENG_Final.pdf
https://www.ilo.org/wcmsp5/groups/public/---ed_dialogue/---sector/documents/publication/wcms_669363.pdf
https://cdn.who.int/media/docs/default-source/health-workforce/youthpaper-final-feb2020_ee459279-a5a6-465f-8859-a0d91b730472.pdf?sfvrsn=a0a4431c_2&download=true

Concerning the statistical analysis, I find these Cronbach’s alphas and the return rate too high. Maybe the responders gave no honest and thoughtful answers but wanted to complete the task fast.

Some remark on the layout:
Please insert a space before an open parenthesis e,g. in line 189, GFI(Goodness.
Please place reference numbers square brackets [ ] in the text.
Please keep all Tables on the same page.
The right side of Figure 1. is not visible, and the Figure and its footnote should be on the same page.

Author Response

Dear Reviewer,

We appreciate the opportunity to revise and resubmit this manuscript. We have read the comments of the reviewer and have revised the manuscript accordingly; we believe our manuscript has benefited immensely from these insightful suggestions for revision.

Point 1.

 To improve this paper, I suggest following the author guide strictly and providing all the appropriate information required in all the sections.
You may want to re-structure the paper to support your conclusions, starting with a problem statement based on a wider literature review. Maybe you will find relevant some of the following documents:

Response 1: Thank you for your comment. We marked the revised part in red

  • 11 to 20 lines, 31 to 33 lines, 41 to 43 lines, 58 to 62 lines, 77 to 79 lines, 85 to 91 lines, 97 to 98 lines

  116 to 117 lines, 136 to 139 lines, 152 to 155 lines, 160 to 163 lines, 182 to 190 lines, 225 to 231 lines,

  284 to 287 lines, 315 to 322 lines, 330 to 336 lines, 374 to 377 lines, 400 to 405 lines.

Point 2.

Concerning the statistical analysis, I find these Cronbach’s alphas and the return rate too high. Maybe the responders gave no honest and thoughtful answers but wanted to complete the task fast

Response 2: Thank you for your comment. We could not evaluate the honesty of the study subject' responses. We'll consider this in my future research.

Point 3.

Some remark on the layout:
Please insert a space before an open parenthesis e,g. in line 189, GFI(Goodness.
Please place reference numbers square brackets [ ] in the text.

Response 3: Thank you for your comment. We marked the revised part in red

Point 4

Please keep all Tables on the same page.
The right side of Figure 1. is not visible, and the Figure and its footnote should be on the same page.

Response 4: Thank you for your comment. We modified it

Reviewer 2 Report

Thank you for allowing me to review "Structural model of earth's retention intention in small and medium-sized hospitals: Based on Herzberg's Motivational Hygiene Story."
I read this well-progressed study interestingly. Thank you for the hard work of the researchers.
To further enhance the value of this study, we present the following opinions.

- Abstract
For this study, the derived variables should be presented. In addition, it seems necessary to describe in detail the results of the research hypothesis adopted. In particular, independent variables that affect the dependent variable should be presented. And the fit of the research model should be excluded. 

- Conceptual basis and hypothetical model 
Previous studies on the influence relationship between variables should be supplemented. Based on this basis, hypotheses should be presented.

-Results
Before verifying the hypothesis, correlation analysis and discriminant validity should be presented.

-Discussion 
"In terms of the general characteristics of small to medium-size hospital numbers, higher age," the general characteristics of this study showed that the composition of people in their 20s and 30s was high. In addition, implications should not be derived based on the general characteristics of the survey subjects.

"The analysis in this study to explain work satisfaction of the small to mid-size hospital nurses resulted in calling, resilience, workplace bullying and nursing work environment, with resilience having the greatest impact on work satisfaction." Looking at Table 3, it is the nursing work environment that has the greatest influence on job satisfaction. 

-Conclusion 
The limitations of this study have not been presented.

I hope there will be good results.

Author Response

Responses to Reviewer 2  Comments

Dear Reviewer,

We appreciate the opportunity to revise and resubmit this manuscript. We have read the comments of the reviewer and have revised the manuscript accordingly; we believe our manuscript has benefited immensely from these insightful suggestions for revision.

Point 1.

Abstract

For this study, the derived variables should be presented. In addition, it seems necessary to describe in detail the results of the research hypothesis adopted. In particular, independent variables that affect the dependent variable should be presented. And the fit of the research model should be excluded.

Response 1: Thank you for your comment. We modified 12 to 20 lines

Point 2. Conceptual basis and hypothetical model

Previous studies on the influence relationship between variables should be supplemented. Based on this basis, hypotheses should be presented.

Response 2: Thank you for your comment. We modified 85 to 103 lines

Point 3. Results

Before verifying the hypothesis, correlation analysis and discriminant validity should be presented.

Response 3: Thank you for your comment. We modified 249 to 267 lines. We added tables 3 and 4

Point 4. Discussion

"In terms of the general characteristics of small to medium-size hospital numbers, higher age," the general characteristics of this study showed that the composition of people in their 20s and 30s was high. In addition, implications should not be derived based on the general characteristics of the survey subjects.

Response 4: Thank you for your comment. We added 318 to 323 lines.

Point 5. Discussion

"The analysis in this study to explain work satisfaction of the small to mid-size hospital nurses resulted in calling, resilience, workplace bullying and nursing work environment, with resilience having the greatest impact on work satisfaction." Looking at Table 3, it is the nursing work environment that has the greatest influence on job satisfaction.

Response 5: Thank you for your comment. We modified 326 lines.

Point 6. Conclusion

The limitations of this study have not been presented.

Response 6: Thank you for your comment. We added 405-406 lines.

Reviewer 3 Report

The authors conducted an interesting study to evaluate which variables were associated with the intention to leave the job in a sample of nurses.

  1. The authors acknowledge that they used a convenience sample. The proportion of female nurses is disproportionate. Authors should indicate in the population description section what the true percentage of females in Korean nurses is, and, if there is a large variation, they should discuss this limitation.
  2. The sample consisted mainly of young nurses. Again, it would be useful to know if the situation of nurses in Korea is of this type. The ageing of the population has also affected nurses, who in most countries of the world have a much higher average age than the one reported here. This difference could limit the extrapolation of the data to other situations.
  3. The authors do not specify when the research was conducted. If it occurred during the pandemic, the authors should warn the reader that the conditions of high pandemic stress may have influenced the intention to leave the workplace, as observed for example in frontline staff with Covid-19 patients [eg.: Magnavita, N.; Soave, P.M.; Antonelli, M. A One-Year Prospective Study of Work-Related Mental Health in the Intensivists of a COVID-19 Hub Hospital. Int. J. Environ. Res. Public Health 2021, 18, 9888. https://doi.org/10.3390/ijerph18189888]
  4. The article is completely missing from the section on weaknesses and strengths

Author Response

Responses to Reviewer 3  Comments

Dear Reviewer,

We appreciate the opportunity to revise and resubmit this manuscript. We have read the comments of the reviewer and have revised the manuscript accordingly; we believe our manuscript has benefited immensely from these insightful suggestions for revision.

Point 1. he authors acknowledge that they used a convenience sample. The proportion of female nurses is disproportionate. Authors should indicate in the population description section what the true percentage of females in Korean nurses is, and, if there is a large variation, they should discuss this limitation.

 Response 1: Thank you for your comment. We added 318 to 320 lines.

Point 2. The sample consisted mainly of young nurses. Again, it would be useful to know if the situation of nurses in Korea is of this type. The ageing of the population has also affected nurses, who in most countries of the world have a much higher average age than the one reported here. This difference could limit the extrapolation of the data to other situations.

 Response 2: Thank you for your comment. We added 320 to 323 lines.

Point 3. The authors do not specify when the research was conducted. If it occurred during the pandemic, the authors should warn the reader that the conditions of high pandemic stress may have influenced the intention to leave the workplace, as observed for example in frontline staff with Covid-19 patients [eg.: Magnavita, N.; Soave, P.M.; Antonelli, M. A One-Year Prospective Study of Work-Related Mental Health in the Intensivists of a COVID-19 Hub Hospital. Int. J. Environ. Res. Public Health 2021, 18, 9888. https://doi.org/10.3390/ijerph18189888]

Response 3: Thank you for your comment. We added 119 to 120 lines. Our study was not affected by COVID-19.

Point 4. The article is completely missing from the section on weaknesses and strengths

Response 4: Thank you for your comment. We added 378 to 380 lines.

Round 2

Reviewer 1 Report

Dear Authors,

It was a pleasure to see the improvement of the submission.

There are still some minor tasks to do, like inserting space before [s, keeping tables together, and polishing the English grammar.

Please correct <Table 3> in line 245.

Bests

Reviewer 2 Report

This study has been well modified and supplemented, making it more valuable than before. I praise your hard work.